# Population Genetic Features of Calving Interval of Holstein-Friesian Cows Bred in Hungary

**DOI:** 10.3390/ani14172513

**Published:** 2024-08-29

**Authors:** Szabolcs Bene, Zsolt Jenő Kőrösi, László Bognár, József Péter Polgár, Ferenc Szabó

**Affiliations:** 1Institute of Animal Sciences, Hungarian University of Agriculture and Life Sciences, Georgikon Campus, Deák Ferenc u. 16, H-8360 Keszthely, Hungary; 2National Association of Hungarian Holstein Friesian Breeders, Lőportár u. 16, H-1134 Budapest, Hungary; 3Albert Kázmér Faculty of Agriculture and Food Sciences, Széchenyi István University, Mosonmagyaróvár, Vár t. 2, H-9200 Győr, Hungary

**Keywords:** calving interval, Holstein-Friesian, heritability, phenotypic and genetic trend

## Abstract

**Simple Summary:**

Calving interval data by the National Association of Hungarian Holstein Friesian Breeders in Hungary were processed. According to our results the relatively low proportion of genetic variances indicates that the selection of Holstein-Friesian cattle for reproductive traits (such as calving interval) may render low magnitudes and long-term responses. Nevertheless, the economic importance of these traits should not be overlooked. Our results on genetic trends, according to which the average calving interval decreased a little during the examined period, suggest that, despite the relatively low genetic determination, it is possible to achieve some improvement in reproductive traits.

**Abstract:**

Calving interval (CI) data (N = 37,263) from 17,319 cows born 2008–2018 in six herds were assessed. The data were made available by the National Association of Hungarian Holstein Friesian Breeders in Hungary. The effects of some genetic and environmental factors, population genetic parameters, breeding value (BV) of sires, and phenotypic and genetic trends of the CI were estimated. The GLM method was used for studying different effects on the CI. BLUP animal model was used for heritability (*h*^2^) and BV estimation. Linear regression analyses were applied for the trend calculation. The mean of the CI was 412.2 ± 2.0 days. The *h*^2^ of the CI proved to be low (0.07 ± 0.01 and 0.08 ± 0.01). There were relatively high differences among the sires in the estimated BV. Based on the phenotypic trend calculation, the CI of cows showed decreasing direction by an average of 1.80 days per year (*R*^2^ = 0.94; *p* < 0.01). In the case of genetic trend calculation, the average BV of sires in the CI has decreased −4.94 and −0.31 days per year (*R*^2^ = 0.91 and 0.41; *p* < 0.01).

## 1. Introduction

In dairy herds, the fertilization period starts approximately 60 days after calving. In terms of milk production, this is the peak period; that is, cows should be inseminated when their daily milk production is at its highest level. In addition to satisfying the maintenance and milk production feed requirement, it would be advisable to pay attention to improving the condition, since the improved nutritional status (“flushing”) is clearly beneficial in terms of fertility. Unfortunately, in practice, optimal nutrition cannot be achieved in many cases, the nutrient requirements of high milk production cannot be met, which can lead to deterioration of the condition or disturbances in the flow of substances. As a result of high-quality milk production, the health of the udder may deteriorate, which may cause an increase in the number of somatic cells, or even mastitis. As a result of all this, the reproductive biological status of the cows is not satisfactory; consequently, it is more difficult for the cows to become pregnant, and the interval between the two calvings becomes longer.

Theoretically, the calving interval (CI) can be shorter than one year. The gestation length is a standard value (285 days overall, 275 days as an average in Holstein-Friesian cows [1]) in cattle and the cows need at least 21, but in practice, a minimum of 42 days, as days open, to be pregnant after calving. Thus, the minimum period between two calvings can be approximately 300–320 days.

There is quite a lot of information about the CI of different cattle breeds in the literature (Table 1).

Among the relevant sources, there are very few where the average value of the CI is less than 365 days [9]. The CI of dairy cattle is usually longer than one year, but many sources report values of over 400 days [8,16]. Some literature sources report an even longer time (430–440 days) between two calvings in dairy herds [17]. In the case of the CI of beef breeds, Yagüe et al. [14] reported 409 days for Rubia Gallega, Silveira et al. [18] reported 465 days for Nellore, and Brzáková et al. [4] reported 370–392 days for Aberdeen Angus and Charolais.

Based on data from most literature sources, the heritability (*h*^2^) and the repeatability (*R*) of the CI trait is very low [19,20]. In the case of the Holstein-Friesian breed, the heritability of the CI trait was 0.03–0.09 [5,13,21]. The relevant literature sources also reported very low *h*^2^ values for beef breeds: 0.13 for Asturiana de los Valles [7], 0.03 for Nellore [22], 0.11–0.18 for mixed genotype [23], 0.06 for Cuban Charolais [24], 0.03 for Hanwoo [9], and 0.08 for Angus and Charolais [4]. The repeatability of the CI trait is 0.01–0.13 [14,25].

Due to the low *h*^2^, environmental factors play a major role in the development of the CI. According to MacGregor and Chasey [26], the calendar date of calving (i.e., month of calving) had a significant effect on the CI. In the studies of Bourdon and Brinks [27], the age at first calving had a significant effect on the CI. Early breeding adversely affected later reproductive characteristics, including the period between calvings [19]. Dunn and Kaltenbach [28] determined that the CI in heifers bred early was almost always longer than the desirable 365 days. The age of the cow [8,14], the fertilization or calving year [18], and the herd [29] have significant effects on the CI.

There is quite a lot of data on the relationship between the CI and the other traits in the literature. The genetic correlation between the age at first calving and the CI was weak [30]. Gutiérrez et al. [31] and Brzáková et al. [4] found strong genetic correlation between the calving difficulty and the CI. In the study of Gutiérrez et al. [7], the genetic correlation between the conformation scoring results and the CI was weak and negative in beef cattle. Nguyen et al. [32] reported a positive relationship between the CI and the somatic cell count.

Some of the existing literature sources report a decrease in the phenotypic and genetic trend of the CI [2,23,33], while others report an increase [8,20,21]. De Rezende et al. [34] found decreasing phenotypic but increasing genetic trends in the Italian Limousin herds.

There is quite a lot of quite diverse and often contradictory information available in the literature on the examination of the CI trait. In addition, the CI is not included in the selection index of the Hungarian Holstein-Friesian; however, it is a very important trait, reflecting the reproduction of dairy cows. Even so, there is very little population genetic information about it. Therefore, the aim of the present study was to determine the *h*^2^ value, the breeding value (BV) of sires, and the phenotypic and genetic trend in the CI trait of Holstein-Friesian cows in Hungary.

## 2. Materials and Methods

From a methodological point of view, our present paper was based on our previous published papers for Holstein-Friesian [35] and for Limousin [36] breeds.

In this manuscript, the methods used to evaluate the data on the CI of the Limousin cows [36] were adapted and further developed for the evaluation of the database of the Holstein-Friesian breed. Therefore, the present manuscript and the previous paper are very similar from a methodological point of view.

In our previous paper [35], we evaluated the CI of the Holstein-Friesian cows, but population genetic parameters, heritability, BV, and trends were not estimated due to the small amount of available data.

All in all, in this manuscript, we applied a method previously used for the CI of beef cattle to a database of dairy cattle. Compared to the results of our previous papers in dairy cattle, the new aspects of this manuscript are the heritability values, BVs, and the phenotypic and genetic trends using large amounts of data.

### 2.1. The Database

The source of the data for the study was the database of the National Association of Hungarian Holstein Friesian Breeders in Hungary. The data of the six biggest large-scale Holstein-Friesian herds were used.

A total of 17,319 cows born between 2008 and 2018 were included in the evaluation. The studied cows were offspring of 842 sires and 13,236 dams (Table 2).

Calvings took place between 2010 and 2022. During this period, a total of 54,582 calving and 37,263 CI data was processed.

The number of female progenies per sire ranged from 5 to 232, with an average of 20.57 offspring per sire. The average CI data per sire was 44.26.

The Kolgomorov–Smirnov test was used to check the normal distribution in the database. The Levene test was used to check the homogeneity of variances.

### 2.2. The Calving Interval Trait

CI was calculated as the difference between calving dates from succeeding parities. In the literature, a great variety of data on the extreme values of the CI trait was found (Table 1). In order to get rid of extreme data, filtering was used, and only CI data of 300–650 days were processed in this study. Lower and higher values were set as missing values.

The year of calving and season of calving was considered to be the start date of the CI.

### 2.3. Examining the Effects of Different Factors

The effect of the different genetic and environmental factors influencing the CI trait was evaluated by GLM (General Linear Model) method (Table 3). Sire was considered as a random effect, while the other examined factors—herd, year of calving, season of calving, and parity of the cow—were considered as fixed effects. The used estimation model was described as follows:(1)$${\overset{\wedge }{y}}_{hijlkm}=\mu +S_{h}+F_{i}+Y_{j}+M_{k}+P_{l}+e_{hijklm}$$(where *ŷ_hijklm_* = CI of cow; born from sire “*h*”, in herd “*i*”, in calving year “*j*”, in calving season “*k*”, and from parity “*l*”; *μ* = mean of all observations; *S_h_* = random effect of sire; *F_i_* = fixed effect of herd; *Y_j_* = fixed effect of year of calving; *M_k_* = fixed effect of season of calving; *P_l_* = fixed effect of parity of cow; and *e_hijklm_* = random error.)

The evaluation of the database was performed with the statistical software package SPSS 27.0 [37].

### 2.4. Estimation of Population Genetic Parameters

To estimate the population genetic parameters, two models—the GLM model [38] and the repeatability BLUP (Best Linear Unbiased Prediction) animal model [39]—were used (Table 3).

The GLM model is presented in Section 2.3. With the GLM method, some population genetic parameters were determined for the CI trait (additive genetic variance, environmental variance, and phenotypic variance). The heritability (*h*^2^) was calculated using the following formula:(2)$$h^{2}=\frac{{\sigma^{2}}_{a}}{{\sigma^{2}}_{a}+{\sigma^{2}}_{e}}=\frac{{\sigma^{2}}_{a}}{{\sigma^{2}}_{p}}$$(where *h*^2^ = heritability; *σ*^2^*_a_* = additive genetic variance; *σ*^2^*_e_* = residual–environmental variance; and *σ*^2^*_p_* = phenotypic variance.)

Using the BLUP models, two matrices were created. One of these was the database matrix and the other was the pedigree matrix. In the BLUP animal model, the same fixed effects were taken into account as in the case of the GLM method. The random effect was the individual (cow). The pedigree matrix of relatives included pedigree data for full sibs, half sibs, sires, dams, and grandparents.

Because a cow can have more CI data, the permanent environmental effect of the cow (PE, as random effect) was also included in the model [40]. Similar to the study of Nagy et al. [41], the used basic repeatability model was as follows:(3)$$y=X_{b}+Z_{a}+W_{pe}+e$$(where “*y*” is the vector of observations; “*b*” is the vector of fixed effects; “*a*” is the vector of random animal effects; “*pe*” is the random vector of permanent environmental effects; “*e*” is the vector of random residual effects; and *X*, *Z*, and *W* are the incidence matrices relating records to fixed, animal, and random permanent environmental effects, respectively).

The repeatability value (*R*) was calculated with the formula as follows [42]:(4)$$R=\frac{{\sigma^{2}}_{a}+{\sigma^{2}}_{pe}}{{\sigma^{2}}_{a}+{\sigma^{2}}_{pe}+{\sigma^{2}}_{e}}$$(where *σ*^2^*_a_* = additive genetic variance; *σ*^2^*_pe_* = permanent environmental effect; and *σ*^2^*_e_* = residual variance.)

For the BLUP animal model, MTDFREML [43] software (https://zzlab.net/MTDFREML/index.html#) was used.

### 2.5. Estimation of Breeding Values

The BV of the sires was also estimated with the GLM method and the BLUP animal model based on the CI trait.

Using the GLM method, the first step was the progeny difference (*EPD*) calculation, as the difference between the average performance of the sire’s progeny group and the average performance of the entire population for the CI trait. In the second step, the BV was calculated as twice the *EPD*. The *EPD* was calculated as follows:(5)$$EPD=(x_{pg}-X_{all})$$(where *x_pg_* = the mean value of the progeny group of the sire and *X_all_* = the mean value of the contemporary offspring population.)

In the case of the BLUP animal model, the model estimated the BV directly. The reliability value (*b*) of the estimated BV was calculated using the following formula:(6)$$b=\sqrt{\frac{n\cdot R\cdot h^{2}}{\frac{1}{R}+(n-1)\cdot h^{2}}}$$(where *b* = reliability value of BV; *n* = number of progenies of sire; *h*^2^ = heritability of CI trait; and *R* = degree of kinship.)

Due to size reasons, BVs are only shown for the 20 sires with the most offspring number.

Based on the estimated BV of the sires in the two different models, two different rankings were established. Similar to some literature sources [44], the effect of the model on the rank of sires was determined by rank correlation calculation [45].

### 2.6. Calculating Phenotypic Trends

During the calculation of the phenotypic trend for the CI, data of cows born in the same year were averaged, and then the mean values were plotted against the year of birth. For fitting function to the resulting set of points, weighted linear regression analysis was used. The dependent variable, (*Y*) was the mean of the CI and the independent (*X*) variable was the birth year of the cow. The values of the constant (*a*), the slope (*b*), and the fit (*R*^2^) and their statistical reliability were also determined.

### 2.7. Estimation of Genetic Trends

The genetic trend of the CI trait—likewise Ostler et al. [46]—was determined from the average BV of animals born in the same year. These were determined in three ways (from the GLM BV of sires, from the BLUP BV of sires, and from the BLUP BV of the entire population born in the same year).

The genetic trend of the CI was evaluated using a linear regression method. The BV of sires as well as the BV of the entire population was averaged annually. The annual mean values were the dependent values, and the appropriate year was the independent value in the used regression method.

Similarly to the phenotypic trend calculation, the value of the constant (*a*), the slope (*b*), and the fit (*R*^2^), as well as their statistical reliability, was determined.

The genetic trends have been estimated for the period between 1997 and 2015 for sires, and between 1997 and 2018 for the entire population.

## 3. Results and Discussion

### 3.1. The Effect of the Environmental Factors

The basic statistical parameters of the CI trait are presented in Table 4. The arithmetic mean of the CI of Holstein-Friesian cows was 409.2 days (SD = 73.1 days, CV = 17.9%). This result was similar to most of the data reported in the literature for dairy cattle [6,8,11]. In contrast, in the case of Holstein-Friesian cows, a longer CI was reported in Ansari-Lari et al. [2] and a shorter CI in Atashi et al. [3]. In the case of beef cattle, we found shorter CI data in the literature [7,12,14] than our results. In the report of Lopez et al. [9] and Brzáková et al. [4], the CI of beef cows was shorter than the CI of dairy cows in the present study.

Subtracting the average gestation length (275 days [1]) of Holstein-Friesian cows from the CI revealed that the average days open of cows in this work was quite long, 134 days. Taking into account the length of the cattle’s estrous cycle (21 days), cows—in an average sense—became pregnant at the fifth estrous cycle. The length of days open in our study was 37 days shorter than what Espionosa et al. [24] reported.

The effect of all environmental factors (herd, year of calving, season of calving, and parity of cow) proved to be significant (*p* < 0.01) on the CI trait (Table 5). The most determining environmental factor was the effect of the season of calving (74.86%). The residual variance was quite small, 0.42%.

The effect of the environmental factors on the CI has been summarized in Table 6. The estimated adjusted overall mean value of the CI by the GLM method proved to be 412.2 ± 2.0 days.

The CI was the shortest in dairy herd number 4 (394.9 ± 2.3 days), which experienced approximately 15 days shorter than that of in the other five herds (413.1–419.5 days). Evaluating the effect of the calving year, the shortest CI (405.2 ± 4.8 days) was observed in 2010 and the longest (421.1 ± 2.2 days) in 2017. The difference between these extremes was 16 days. The effect of herd and calving year was similar to the data found in most literature sources [18,29].

The CI of autumn-calved cows (400.7 ± 2.0 days) was on average 27 days shorter than that of spring-calved counterparts (427.8 ± 2.2 days). During the examination of the effect of the calving season, Kanuya and Greve [17] experienced something similar.

The effect of the parity of dam was smaller than expected and in the opposite direction. The longest time (414–416 days) between two calvings was calculated for cows of prime age (parity of dam 3–7). The shortest CI was observed after the first calving (406.4 ± 0.9 days). This result differed from the data of Nieuwhof et al. [10], Hare et al. [8], and Yagüe et al. [14], who observed an increase in the CI with increasing maternal age.

### 3.2. Population Genetic Parameters

The *h*^2^ value (0.07 ± 0.01 and 0.08 ± 0.01 with two models) of the CI trait proved to be very small (Table 7). The heritability of the CI trait of Holstein-Friesian cows was similar to the data from the relevant literature sources [5,13,21]. In contrast, some literature sources [4,23,24] published slightly higher *h*^2^ values for the CI trait.

There was no difference between the population genetic parameters estimated using the two different models. Similarly to our previous results [36,38], it seems that in special cases (one breed, only cows, etc.) reliable results can also be obtained with simpler models.

Using the BLUP animal model, the value of a permanent environmental effect (*σ*^2^*_pe_* = 26.33) and the ratio of the permanent environmental variance to the phenotypic variance (*c*^2^ = 0.01 ± 0.01) was very low. Therefore, the repeatability value of the CI trait was also estimated to be very small (*R* = 0.09 ± 0.01). The relevant literature sources [9,25] also reported very low repeatability value for the CI trait.

### 3.3. The Effect of Sire and the Breeding Values

Using the GLM method, considerable differences were found between the mean values of the CI of the progeny groups of different sires (Table 8). The difference between the shortest (393.6 ± 6.3 days; sire code number 18) and the longest (441.6 ± 4.9 days; sire code number 1) female progeny groups was 48.0 days as an average CI. Since the difference between the progeny groups was quite large, large differences between the BV of the CI of the sires were also found. Accordingly, the difference between the smallest (−37.2 ± 6.3 days) and the largest (+58.7 ± 5.1 days) estimated BV was 95.9 days.

Using the BLUP animal model, the difference between the two extremes of BV (−22.7 days and +21.8 days) was 44.5 days. Generally, the breeding values estimated with the BLUP animal model were smaller than those estimated with the GLM method. In the case of the BLUP animal model, the difference between the extreme values was smaller than in the case of the GLM method.

In the relevant literature sources, numerically displayed BVs for the CI cannot be found.

Between the ranking of sires (with the GLM method and the BLUP animal model) medium, positive rank correlation (r_rank_ = +0.60; *p* < 0.01) value was estimated. In the study of Núnez-Dominguez et al. [44], the rank correlation value was higher between the ranks of sires. During our previous studies [36,38], a closer correlation between the ranking of sires—based on the BV estimated with different models—was found.

### 3.4. Phenotypic and Genetic Trends

The computed phenotypic and genetic trends are presented in Table 9 and Figure 1. The phenotypic trend was based on the birth year of the cows. The genetic trends were based on the birth year of the sires, or the birth year of the entire population.

Based on the phenotypic trend calculation, the CI of Holstein-Friesian cows decreased by an average of 1.80 days per year (*b* = −1.80 ± 0.15 days; *p* < 0.01). The fit of the phenotypic trend (*R*^2^ = 0.94; *p* < 0.01) was very high and significant. This indicates a remarkable decrease in the CI in the studied Holstein-Friesian population.

Based on the birth year of the sires, the genetic trend by the GLM method showed −4.94 days decrease per year (*b* = −4.94 ± 0.37 days; *p* < 0.01). The fitting value was high (*R*^2^ = 0.91; *p* < 0.01). By the BLUP animal model, the slope (*b* = −0.31 ± 0.09 days; *p* < 0.01) and the fitting (*R*^2^ = 0.41; *p* < 0.01) value were very similar in direction, but smaller than the GLM method.

Based on the estimated BV of the entire population, the genetic trend of the CI was slightly decreasing (*b* = −0.09 ± 0.03 days; *p* < 0.01), but it was statistically proven (*R*^2^ = 0.33; *p* < 0.01).

Similarly to some literature data [2,23,33], a small decrease in the genetic trend of the CI was observed. Older sources [8,21] typically reported an increase in the phenotypic and genetic trend of the CI trait.

## 4. Conclusions

The CI is a very important trait in the breeding of dairy cattle as it is a very good indicator for characterizing the reproductive state of the cow herd. If the CI is extended, it can usually lead to deterioration in the results of reproduction, rearing, and milk production. For this reason, the goal of breeders is to reduce the CI, which they try to achieve with reproductive biological methods as well as breeding methods. The latter can be successful, if information is available on the population genetic parameters of the CI trait, as well as the estimated BV of the bulls in the CI trait.

According to our results, the relatively low proportion of genetic variances indicates that the selection of Holstein-Friesian cattle for reproductive traits (such as CI) may render low magnitudes and long-term responses. Nevertheless, the economic importance of these traits should not be overlooked.

Based on our results, it seems worth considering the possibility of incorporating CI into the selection index.

Our results on genetic trends, according to which the average CI decreased a little during the examined period, suggest that, despite the relatively low genetic determination, it is possible to achieve some improvement in reproductive traits.

## Figures and Tables

**Figure 1 animals-14-02513-f001:**
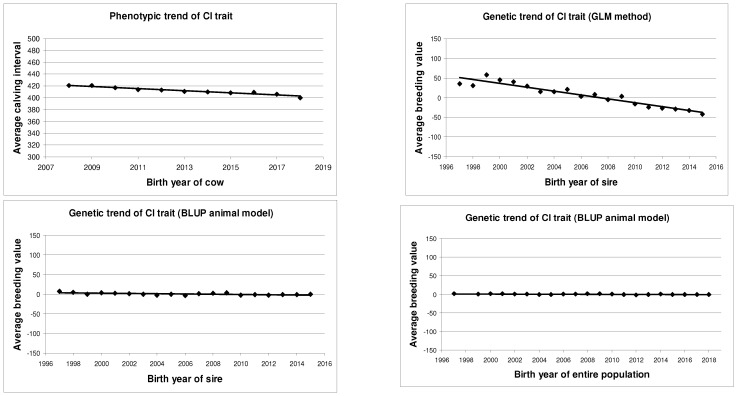
Phenotypic and genetic trends of CI trait of Holstein-Friesian cows.

**Table 1 animals-14-02513-t001:** The mean and the range of the CI trait in the literature.

Breed	Calving Interval		Source
	Range (Days)	Mean (Days)	
HOL	260–750	389–435	Ansari-Lari et al. [2]
HOL	270–700	395	Atashi et al. [3]
ABA, CHA	290–500	370–392	Brzáková et al. [4]
SIM	300–700	398	Cesarani et al. [5]
HOL	330–750	414	Fiedlerová et al. [6]
ASV	290–630	488	Gutiérrez et al. [7]
AYR, BRS, GUE, HOL, JER	270–650	390–407	Hare et al. [8]
HAN	300–600	363	Lopez et al. [9]
AYR, BRS, GUE, HOL, JER	270–650	393–405	Nieuwhof et al. [10]
HOL, JER, CRO	310–600	413	Schambow et al. [11]
HAN	300–600	399	Shin et al. [12]
HOL	<700	384–399	Short et al. [13]
RUB	254–629	409	Yagüe et al. [14]
HOL	324–586	360–512	Wang et al. [15]

ABA = Aberdeen Angus; ASV = Asturiana de los Valles; AYR = Ayrshire; BRS = Brown Swiss; CHA = Charolais; CRO = crossbred; GUE = Guernsey; HAN = Hanwoo; HOL = Holstein-Friesian; JER = Jersey; RUB = Rubia Gallega; SIM = Simmental.

**Table 2 animals-14-02513-t002:** The structure of the evaluated database for the Holstein-Friesian population.

Starting Parameters	Used Database
Number of cows	17,319
Birth date of cows	2008–2018
Number of calvings	54,582
Period of calving	2010–2022
Number of calvings per cow	3.15
Number of calving interval data (N)	37,263
Average calving interval data per cow	2.15
Number of herds	6
Number of the examined sires (sire of cow)	842
Birth date of sires	1997–2015
Average number of female progeny (cow) per sire	20.57
Average calving interval data per sire	44.26
Number of the examined dams (dam of cow)	13,236
Birth date of dams	1996–2017

**Table 3 animals-14-02513-t003:** The applied models for the estimations.

Type of Model	GLM Method	BLUP Animal Model
Random effects		
-sire (sire of cow)	+	−
-animal (cow)	−	+
-permanent environmental effect	−	+
Fixed effects		
-herd	+	+
-year of calving	+	+
-season of calving	+	+
-parity of cow (age of cow at calving)	+	+
Pedigree matrix		
-sire (sire of cow)	−	+
-dam (dam of cow)	−	+
-full sibs, half sibs	−	+
-grandparents	−	+
Examined trait		
-calving interval	+	+

+ = the model includes this effect; − = the model does not include this effect.

**Table 4 animals-14-02513-t004:** Descriptive statistics of CI trait of Holstein-Friesian cows.

Parameters	Calving Interval
N	37,263
Mean (days)	409.2
Standard error (SE) (days)	0.4
Standard deviation (SD) (days)	73.1
Coefficient of variation (cv%)	17.9
Median (days)	389.0
Range (days)	350
Minimum (days) *	300
Maximum (days) *	650
Kolgomorov–Smirnov test ^#^ (*p*)	0.00

* according to Table 1; ^#^ if *p* > 0.05, the normal distribution is confirmed.

**Table 5 animals-14-02513-t005:** The effect of different factors on the CI trait of Holstein-Friesian cows.

Trait	Classes	Calving Interval		
Factor		The Effect and Rate of Factors in Phenotype		Levene-Test *
		*p*	%	*p_L_*
Sire of cow	842	<0.01	0.77	0.00
Herd	6	<0.01	16.49	0.00
Year of calving	13	<0.01	2.07	0.00
Season of calving	4	<0.01	74.86	0.00
Parity of cow	7	<0.01	5.39	0.00
Residual	-	-	0.42	-
Total	-	-	100.00	-

* if *p_L_* > 0.05, the homogeneity is confirmed.

**Table 6 animals-14-02513-t006:** The effect of the environmental factors on the CI trait of Holstein-Friesian cows.

Trait	N	Calving Interval (Days)	
Adjusted Overall Mean (±SE)	37,263	412.2 ± 2.0	
Environmental Factors		Mean ± SE	Deviation from Overall Mean
Herd (Code)			
1	4727	413.1 ± 2.6	+0.8
2	9918	419.5 ± 2.1	+7.2
3	4191	413.7 ± 2.6	+1.5
4	7846	394.9 ± 2.3	−17.4
5	7846	416.8 ± 2.5	+4.6
6	2735	415.5 ± 2.7	+3.3
Year of Calving			
2010	1160	405.2 ± 4.8	−7.0
2011	2095	407.4 ± 4.2	−4.9
2012	2770	409.0 ± 3.7	−3.2
2013	3101	409.9 ± 3.4	−2.3
2014	3149	408.1 ± 3.0	−4.1
2015	3318	413.8 ± 2.7	+1.6
2016	3467	413.6 ± 2.4	+1.4
2017	3450	421.1 ± 2.2	+8.9
2018	3629	417.2 ± 2.0	+5.0
2019	3846	414.7 ± 2.0	+2.4
2020	3811	416.7 ± 2.1	+4.4
2021	2274	413.1 ± 2.5	+0.8
2022	1193	409.3 ± 3.1	−2.9
Season of Calving			
winter	9255	411.7 ± 2.1	−0.5
spring	6664	427.8 ± 2.2	+15.6
summer	10,384	408.8 ± 2.0	−3.5
autumn	10,960	400.7 ± 2.0	−11.6
Parity of Cow			
2	17,139	406.4 ± 0.9	−5.8
3	10,897	414.9 ± 0.9	+2.7
4	5797	414.3 ± 1.3	+2.1
5	2382	411.5 ± 2.0	−0.7
6	738	414.2 ± 3.2	+2.0
7	239	416.8 ± 5.2	+4.5
8	71	407.5 ± 8.9	−4.7

**Table 7 animals-14-02513-t007:** Population genetic parameters of the CI trait of Holstein-Friesian cows.

Parameters	Calving Interval	
	GLM Method	BLUP Animal Model
*σ* ^2^ * _a_ *	376.78	411.48
*σ* ^2^ * _pe_ *	-	26.33
*σ* ^2^ * _e_ *	5003.87	4600.93
*σ* ^2^ * _p_ *	5380.65	5038.73
*h* ^2^	0.07 ± 0.01	0.08 ± 0.01
*c* ^2^	-	0.01 ± 0.01
*e* ^2^	-	0.91 ± 0.01
*R*	-	0.09 ± 0.01

*σ*^2^*_a_* = additive direct genetic variance; *σ*^2^*_pe_* = permanent environmental effect; *σ*^2^*_e_* = residual variance; *σ*^2^*_p_* = phenotypic variance; *h*^2^ = heritability; *c*^2^ = the ratio of the permanent environmental variance to the phenotypic variance; *e*^2^ = the ratio of the residual variance to the phenotypic variance; *R* = repeatability.

**Table 8 animals-14-02513-t008:** The effect of sire and breeding values on the CI trait of Holstein-Friesian cows.

Trait	N	Calving Interval (Days)				
Adjusted Overall Mean (±SE)	37,263	412.2 ± 2.0				
Sire of Cow (Code Number)		GLM Method			BLUP Animal Model	
		Mean of Progeny Group (±SE)	*EPD*	BV ± SE	BV	*b*
1	236	441.6 ± 4.9	+29.4	+58.7 ± 5.1	+21.8	0.7
2	204	430.5 ± 5.3	+18.3	+36.5 ±5.7	+2.1	0.7
3	280	415.5 ± 4.5	+3.2	+6.4 ± 4.4	−1.0	0.7
4	245	431.5 ± 4.9	+19.3	+38.6 ± 4.8	+6.1	0.7
5	560	421.6 ± 3.6	+9.4	+18.8 ± 3.9	+21.4	0.7
6	219	429.2 ± 5.1	+17.0	+34.0 ± 5.7	+6.3	0.7
7	224	425.4 ± 5.0	+13.2	+26.4 ± 5.0	−2.0	0.7
8	223	415.1 ± 5.1	+2.9	+5.8 ± 5.3	−5.7	0.7
9	233	408.3 ± 5.0	−3.9	−7.9 ± 5.0	−21.4	0.7
10	231	403.7 ± 5.0	−8.6	−17.1 ± 5.5	−14.8	0.7
11	465	401.4 ± 3.7	−10.8	−21.6 ± 3.9	−22.7	0.7
12	208	413.1 ± 5.3	+0.9	+1.7 ± 5.5	−8.1	0.7
13	212	411.8 ± 5.2	−0.4	−0.9 ± 5.9	−3.1	0.7
14	253	400.9 ± 4.9	−11.4	−22.7 ± 4.9	−21.4	0.7
15	208	408.2 ± 5.6	−4.0	−8.0 ± 6.0	+11.8	0.7
16	199	401.7 ± 5.8	−10.5	−21.1 ± 6.2	−13.9	0.7
17	218	410.0 ± 5.8	−2.2	−4.5 ± 5.8	+4.9	0.7
18	198	393.6 ± 6.3	−18.6	−37.2 ± 6.3	−13.5	0.7
19	383	399.9 ± 4.9	−12.4	−24.8 ± 4.7	−2.5	0.7
20	254	400.8 ± 5.9	−11.4	−22.8 ± 5.8	−1.9	0.7
r_rank_ (BV_GLM_ − BV_Blup_)			0.60 (*p* < 0.01)			

N = number of progenies; *EPD* = estimated progeny difference; BV = breeding value; *b* = reliability value of BV; BV_GLM_ = breeding value with GLM method; BV_Blup_ = breeding value with BLUP animal model; r_rank_ = Spearman rank-correlation value.

**Table 9 animals-14-02513-t009:** Phenotypic and genetic trends in the CI trait of Holstein-Friesian cows.

Trend	Y	Slope			Intercept			Fitting	
		bX			a				
		*b*	SE	*p*	*a*	SE	*p*	*R* ^2^	*p*
Phenotypic (X_1_)	aCI	−1.80	0.15	<0.01	4039.68	300.28	<0.01	0.94	<0.01
Genetic (X_2_)									
−BV_GLM_ of sires	aCI^BV^	−4.94	0.37	<0.01	9914.49	736.70	<0.01	0.91	<0.01
−BV_Blup_ of sires	aCI^BV^	−0.31	0.09	<0.01	615.26	179.03	<0.01	0.41	<0.01
Genetic (X_3_)									
−BV_Blup_ of all	aCI^BV^	−0.09	0.03	<0.01	175.44	57.83	<0.01	0.33	<0.01

*b* = slope; *a* = constant; *R*^2^ = fitting; X_1_ = birth year of cow; X_2_ = birth year of sires; X_3_ = birth year of entire population; aCI = average calving interval (days); aCI^BV^ = average breeding value in calving interval (days); BV_GLM_ = breeding value with GLM method; BV_Blup_ = breeding value with BLUP animal model.

## Data Availability

The data presented in this study are available on request from the National Association of Hungarian Holstein Friesian Breeders.

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
