# Peer review of "Population Genetic Features of Calving Interval of Holstein-Friesian Cows Bred in Hungary"

_animals, 2024, doi:10.3390/ani14172513_

Round 1
Reviewer 1 Report
Comments and Suggestions for Authors
Comments
The rationale of this analysis has to be worked out. The breeding program and routinely EBVs have to be described. The analysis has to be in embedded in the context of the breeding program.
Conclusions have to be amended. Breeding for fertitilty and longevity is applied in Holsteins. Is CI a trait in the total merit index in the Hungarian Friesian cattle. Would also expect to see other fertility and longevity traits besides CI.
Objectives are missing in the Introduction. Please also explain why you analyse CI and not further traits reflecting longevity and insemination success in dairy cows.
Model: the standard model in animal breeding are animal models with deep numerator relationship matrices. Why do you use a model with sire as a fixed effect. With GLM, it is not possible to parameterize a random effect. You should use MIXED or GLIMMIX.
BLUP-model is unclear. The random effects were animal, dam and residual, but not fullsibs, sire, ...
Did you estimate additive genetic variance.
σ2d = genetic variance >> σ2a additive genetic variance
BV >> EBV
I do not think that you can estimate breeding values with a GLM. EDP does not weight by effective daugthers nor for heritability. Please correct your formula.
Table 9: estimates of intercepts with their SE: look very strange.
Comments on the Quality of English Language
No comments
Author Response
The line numbers indicated in the response refer to the corrected version of the manuscript!
Comment 1: The rationale of this analysis has to be worked out. The breeding program and routinely EBVs have to be described. The analysis has to be in embedded in the context of the breeding program.
Response 1: The calving interval trait is not directly mentioned in the Hungarian Holstein-Friesian breeding program, in breeding goals and in selection index. The breeding association regularly measures the calving interval, but as a result of the above, we have very little population genetic information about the trait. Therefore, we chose as the primary goal of our manuscript the evaluation of the calving interval trait.
Accordingly, we added a sentence to the objectives of our manuscript: “In addition, the CI trait is not included in the selection index of the Hungarian Holstein-Friesian, however it is a very important trait, reflecting the reproduction of dairy cows. Even so, there is very little population genetic information about it.” (line 90-93).
Comment 2: Conclusions have to be amended. Breeding for fertility and longevity is applied in Holsteins. Is CI a trait in the total merit index in the Hungarian Friesian cattle. Would also expect to see other fertility and longevity traits besides CI.
Response 2: Among the mentioned traits, only longevity is included in the Hungarian Holstein-Friesian selection index. The fertility and the calving interval are not part of the index. The examination of reproductive traits is of course part of the performance testing, but population genetic parameters are available usually only for the longevity trait. In the conclusions chapter, we wanted to emphasize the importance of the CI trait, as CI also belongs to the traits related to reproduction. In our opinion, it would be worthwhile to include the CI trait in the selection index.
To emphasize this, we inserted a few sentences at the end of the conclusions chapter: “Based on our results, it seems that the CI trait may be worth incorporating into the selection index.” (line 329-330).
Comment 3: Objectives are missing in the Introduction. Please also explain why you analyse CI and not further traits reflecting longevity and insemination success in dairy cows.
Response 3: The supplemented objectives of our work are summarized in detail at the end of the literature chapter (line 89-95).
Comment 4: Model: the standard model in animal breeding are animal models with deep numerator relationship matrices. Why do you use a model with sire as a fixed effect. With GLM, it is not possible to parameterize a random effect. You should use MIXED or GLIMMIX.
Response 4: Thank you very much for your comment, the most accurate results can be obtained by the animal model during animal breeding calculations. In addition, we also presented our results obtained with a simple GLM procedure in our manuscript. The reason for this was that during many of our previous works, where we evaluated special databases (only one breed, only cows, etc.), the results calculated with the GLM procedure and the animal model did not differ significantly from each other. We would like to draw attention to the usability of the simpler method, although the comparison of the models was not the purpose of our present work.
We described in the material and method chapter (line 132-135), and displayed in Table 3 (line 158) that the sire's effect was included as a random effect in the model in the case of the GLM method.
With the GLM procedure we used, we were able to treat the effect of the sire as a random effect. (SPSS 27.0; Analyze/General linear model)
The other Reviewers had no substantive objections to the GLM method. Nonetheless, if the respected Reviewer insists on it, the results obtained with the GLM method will be removed from the manuscript.
Comment 5: BLUP-model is unclear. The random effects were animal, dam and residual, but not fullsibs, sire, ...
Response 5: We apologize, there was a typo in Table 3, which was corrected (line 158). The detailed description of the used animal model, the used pedigree and data matrices, fixed and random effects can be found in Table 3 and in the 2.4 chapter (line 143-171).
Comment 6: Did you estimate additive genetic variance; σ2d = genetic variance >> σ2a additive genetic variance; BV >> EBV
Response 6: Yes, the calculated σ2d genetic variance value is the same as the σ2a additive genetic variance value (corrected everywhere in the manuscript; line 148, 151, 169, 267).
No, the BV was calculated as twice the EPD (line 177-178).
Comment 7: I do not think that you can estimate breeding values with a GLM. EDP does not weight by effective daughters nor for heritability. Please correct your formula.
Response 7: In short, the breeding value expresses how much the average performance of the offspring of an individual differs from the average performance of the entire population. During our work, we calculated the EPD values ​​according to this guideline. The applied GLM method did not really estimate breeding values, but it determined the average performance of the offspring groups of the sires and the average performance of the entire population. From these, the EPD can be determined by a simple subtraction, followed by a multiplication of the breeding value.
Comment 8: Table 9: estimates of intercepts with their SE: look very strange.
Response 8: The results in Table 9 have been recalculated. Despite their strangeness, the values are correct.
Point 1: Minor editing of English language required
Response to point 1: The textual parts of the manuscript were reviewed repeatedly, and the English translation was corrected in many places. We tried to use the grammar formulas used in scientific papers.
Thank you very much for the review. We tried to answer all reviewers' questions and correct the manuscript according to the review.

Reviewer 2 Report
Comments and Suggestions for Authors
The data analysis in this study is complete and the conclusions are credible. In this study, calving interval data from 12 Holstein Friesland cattle were processed across Hungary, and the relatively low proportion of genetic variation suggests that selection of reproductive traits (e.g. calving interval) in Holstein-Friesland cattle may have low amplitude and long-term responses. The genetic trend results showed a slight decrease in the average calving interval during the study period, suggesting that improvement in reproductive traits is still possible despite relatively low genetic decisions.
However, there are some small aspects that could be improved, such as:
1. The purpose and significance of the study are not indicated in the abstract.
2. Materials and Methods The beef cattle method is used to study dairy cows, although the amount of data is increased in comparison, but whether it will have an impact on the results, whether other studies have been implemented, please list the support.
3. The results are presented in a single form, and there is little discussion in the discussion section, for example, when describing the influence of different factors, only the results are elaborated, why these factors affect the trait, and it is recommended to modify and increase the length of the discussion.
4. The references are far away, and it is recommended to add more literature from the last three to five years.
Comments on the Quality of English Language
.
Author Response
The line numbers indicated in the response refer to the corrected version of the manuscript!
Comment 0: The data analysis in this study is complete and the conclusions are credible. In this study, calving interval data from 12 Holstein Friesland cattle were processed across Hungary, and the relatively low proportion of genetic variation suggests that selection of reproductive traits (e.g. calving interval) in Holstein-Friesland cattle may have low amplitude and long-term responses. The genetic trend results showed a slight decrease in the average calving interval during the study period, suggesting that improvement in reproductive traits is still possible despite relatively low genetic decisions.
However, there are some small aspects that could be improved, such as:
Comment 1: The purpose and significance of the study are not indicated in the abstract.
Response 1: The Animals journal requirements for both the simple summary and the summary are very strict. The length of these parts can be 200-200 words. Due to space limitations, the original summary had to be greatly shortened, so there was no "space" left in the summary to present the significance of the study.
In addition, the aim of our study was explained in detail at the end of the literature chapter (line 89-95).
Comment 2: Materials and Methods The beef cattle method is used to study dairy cows, although the amount of data is increased in comparison, but whether it will have an impact on the results, whether other studies have been implemented, please list the support.
Response 2: We adopted only the statistical methods from previous research with beef cattle. When compiling the models, we took into account similar aspects and used the same software. We also kept the format of the tables and figures. This was the total similarity with previous studies. No other type of or new calculation or examination was carried out.
In the case of the Holstein-Friesian, we had a much larger database available. Accordingly, the results also differed significantly from the previous ones.
Comment 3: The results are presented in a single form, and there is little discussion in the discussion section, for example, when describing the influence of different factors, only the results are elaborated, why these factors affect the trait, and it is recommended to modify and increase the length of the discussion.
Response 3: Thank you very much for the suggestion. In the results chapter, we mainly focused on the presentation of our calculated data and their comparison with literature data. We tried to present the entire chapter in an "objective" form, avoiding all guesswork and comments not based on our results.
We have written a paragraph about each investigated factor (line 211-251).
Comment 4: The references are far away, and it is recommended to add more literature from the last three to five years.
Response 4: Of the used sources, 10 are from the last 5 years. We have tried to compile the literature section in such a way that, in addition to the results of the past period, the data of older papers are also presented. Thus, it became possible to follow the change of the calving interval trait, which was a great help in the evaluation of trends.
We used 56 literature sources for our manuscript. Recently, we have found relatively few literature sources on the examination of the calving interval trait. If the respected Reviewer agrees, in consequence of these we did not add any additional resources to the manuscript.
If the Reviewer deems it absolutely necessary to expand the literature section, we would be happy to receive suggested source works.
Point 1: Minor editing of English language required
Response to point 1: The textual parts of the manuscript were reviewed repeatedly, and the English translation was corrected in many places. We tried to use the grammar formulas used in scientific papers.
Thank you very much for the review. We tried to answer all reviewers' questions and correct the manuscript according to the review.

Reviewer 3 Report
Comments and Suggestions for Authors
The introduction gives enough information on the background of the investigated trait.
Please check the English throughout the paper. I pointed out few problems, but you need to carefully check the whole paper.
Line 23: “moreover” is not needed here
Line 54, Table 1: please remove the line below Ansari-Lari et al.; please check “Cesarini” since in reference you have “Cesarani”
Line 72: “between the calendar date of calving and the CI” what do you mean?
Line 77: “are significant effect” should be “are significant effects”
Line 79: “The genetic correlation between the age at first calving and CI was loose” we do not use loose quite often for genetic correlation, maybe it is better to use “weak”
Lines 81-82: “Between the conformation scoring results and CI were loose negative genetic correlation” this sentence is not quite English
Lines 93-105: these lines are not scientifically sound. Please clarify and the improve this part. Did you consider in this manuscript the same data already published? If yes, you cannot have this paper published, if not these lines make no sense.
Lines 128: How did you perform the GLM with the random effect of the sire? Which function or software did you use?
Lines 141-143: What is “genetic variance among the progeny groups”? Did you mean the genetic variance associated with the random effect of sire included in the GLM?
Line 148: “the model include this effect” should be “includes”
Lines 153-154: “Because a cow can have more CI data, the maternal permanent environmental effect (as random effect) was also included in the model” This sentence makes no sense. The maternal permanent environment (MPE) does not account for the repeated calving intervals of a cow. If a cow has repeated measurement, i.e., repeated calving intervals, you must use the permanent environment (PE), because the animals have repeated measurements not their dam. The MPE is used to model different situations.
Line 162: Sigma(d) is already associated with “genetic variance among the progeny groups”; please consider using Sigma(a) as most manuscripts about quantitative genetics.
Line 174: “the model communicated the values of BV directly” a model doesn’t communicate…
Lines 222-223: are you sure about this result? 0.42% is very small.
Lines 253: “the - the ratio”
Line 257, Table 7: based on your variance components, the repeatability of your trait should be 0.09, not 0.08 as you wrote.
Lines 271-273: please check this sentence, is not clear
Lines 276-278: the wording of this sentence is inverted
Line 281, Table 8: what does the “b” (last column) mean here?
Line 286: you do not “estimate” a phenotypic trend, you *compute* a phenotypic trend, since you simply averaged the CI values within year of birth
Lines 299-300: “Similarly to some literature sources [5,26,39] results,” this makes no sense
Line 306, Figure 1: are you sure about these genetic trends?
Comments on the Quality of English Language
You must check carefully the English language throughout the manuscript.
Author Response
The line numbers indicated in the response refer to the corrected version of the manuscript.
Comment 0: The introduction gives enough information on the background of the investigated trait.
Please check the English throughout the paper. I pointed out few problems, but you need to carefully check the whole paper.
Comment 1: Line 23: “moreover” is not needed here
Response 1: Thank you for your comment, the word “moreover” has been deleted (line 23).
Comment 2: Line 54, Table 1: please remove the line below Ansari-Lari et al.; please check “Cesarini” since in reference you have “Cesarani”
Response 2: Thank you for your comment, the line has been deleted. The name was checked, the correct name is "Cesarani". We have corrected it everywhere in the manuscript (line 54).
Comment 3: Line 72: “between the calendar date of calving and the CI” what do you mean?
Response 3: This designation comes from a paper by MacGregor and Chasey (1999), who wrote the follows:
“A one day delay in previous calving date, resulted in a 0.72 day decrease in calving interval, and a 0.27 day increase in calving date.” (https://doi.org/10.1016/S0301-6226(98)00158-4)
The sentence has been replaced by the following: “According to MacGregor and Chasey [31], the calendar date of calving (i. e. month of calving) had a significant effect on the CI.” (line 71-72).
Comment 4: Line 77: “are significant effect” should be “are significant effects”
Response 4: Thank you very much. The grammatical error has been corrected (line 77).
Comment 5: Line 79: “The genetic correlation between the age at first calving and CI was loose” we do not use loose quite often for genetic correlation, maybe it is better to use “weak”
Response 5: Thanks for the suggestion. We changed the word "loose" to "weak" (line 79).
Comment 6: Lines 81-82: “Between the conformation scoring results and CI were loose negative genetic correlation” this sentence is not quite English
Response 6: Thank you for the grammatical comment. The named sentence has been replaced by the following: “In study of Gutiérrez et al. [10], the genetic correlation between the conformation scoring results and CI was weak and negative in beef cattle.” (line 81-83).
Comment 7: Lines 93-105: these lines are not scientifically sound. Please clarify and the improve this part. Did you consider in this manuscript the same data already published? If yes, you cannot have this paper published, if not these lines make no sense.
Response 7: In this paper, we used a similar data analysis procedure as in previous manuscript (https://doi.org/10.1556/004.2022.00008). The previous and the present work are similar only in the used statistical procedures and the used software. Apart from that, there are no other similarities between the two papers.
Originally, the named section was not part of the manuscript. This part was included in the manuscript during a previous correction work, at the special request of the Editor. If the respected Reviewer and the Editor see it as better, we will remove this part from the manuscript later.
Comment 8: Lines 128: How did you perform the GLM with the random effect of the sire? Which function or software did you use?
Response 8: The SPSS 27.0 software (Analyze/General linear model) was used for the calculation. With this procedure, fixed and random effects, covariates, etc. can be built into the model.
Comment 9: Lines 141-143: What is “genetic variance among the progeny groups”? Did you mean the genetic variance associated with the random effect of sire included in the GLM?
Response 9: Yes. By the name "genetic variance among the progeny groups" we mean the “genetic variance”, or from another approach, the “additive genetic variance”. At the request of the other Reviewer, we changed this to "additive direct genetic variance" uniformly throughout the manuscript, which is denoted by sigma(a) (σ2a) (line 148, 151, 169, 267).
Comment 10: Line 148: “the model include this effect” should be “includes”
Response 10: Thank you for the comment, the grammatical error has been corrected (line 159).
Comment 11: Lines 153-154: “Because a cow can have more CI data, the maternal permanent environmental effect (as random effect) was also included in the model” This sentence makes no sense. The maternal permanent environment (MPE) does not account for the repeated calving intervals of a cow. If a cow has repeated measurement, i.e., repeated calving intervals, you must use the permanent environment (PE), because the animals have repeated measurements not their dam. The MPE is used to model different situations.
Response 11: Thank you very much for the useful comment. We used and interpreted permanent environment in this form, but unfortunately translated it incorrectly into English. In the entire manuscript, the word "maternal" has been deleted from the beginning of the term (line 158, 160, 169, 261)
Comment 12: Line 162: Sigma(d) is already associated with “genetic variance among the progeny groups”; please consider using Sigma(a) as most manuscripts about quantitative genetics.
Response 12: Thanks for the suggestion. We answered this in comment 9.
Comment 13: Line 174: “the model communicated the values of BV directly” a model doesn’t communicate…
Response 13: Thank you for the grammar and word usage comment. The sentence has been corrected to the following: "In the case of BLUP animal model, the model estimated the BV directly." (line 181-182).
Comment 14: Lines 222-223: are you sure about this result? 0.42% is very small.
Response 14: The rate of factors in phenotype was calculated from F values in ANOVA table. We performed the relevant calculations again. The 0.42 value seems good. Based on our previous research, it seems that when evaluating databases with a large number, the residual is usually small.
Comment 15: Lines 253: “the - the ratio”
Response 15: Thank you very much, the repetition has been deleted (line 262).
Comment 16: Line 257, Table 7: based on your variance components, the repeatability of your trait should be 0.09, not 0.08 as you wrote.
Response 16: Thank you, really. The typo has been corrected (line 264, 266).
Comment 17: Lines 271-273: please check this sentence, is not clear
Response 17: The named sentence has been corrected to the following: “Generally, the breeding values estimated with BLUP animal model were smaller than estimated with GLM method. In case of BLUP animal model, the difference between the extreme values was smaller than in case of GLM method.” (line 280-283).
Comment 18: Lines 276-278: the wording of this sentence is inverted
Response 18: Thank you for the grammatical comment. The sentence has been transformed as follows: “Between the ranking of sires (with in GLM method and BLUP animal model) medium, positive rank correlation (rrank = +0.60; p <0.01) value was estimated.” (line 286-287).
Comment 19: Line 281, Table 8: what does the “b” (last column) mean here?
Response 19: The “b” is reliability value. The method of calculation was described in detail in chapter 2.5 (line 181-184).
Comment 20: Line 286: you do not “estimate” a phenotypic trend, you *compute* a phenotypic trend, since you simply averaged the CI values within year of birth
Response 20: Thank you very much for the numerous comments related to grammar and English word usage. We replaced the word "estimated" with "computed" in the sentence (line 292).
Comment 21: Lines 299-300: “Similarly to some literature sources [5,26,39] results,” this makes no sense
Response 21: Thank you, really. The beginning of the sentence has been changed to: “Similarly to some literature data [5,26,39],” (line 313-314).
Comment 22: Line 306, Figure 1: are you sure about these genetic trends?
Response 22: Based on our calculations, we obtained these genetic trends as a result. Perhaps the slope of the genetic trend determined by the GLM method was somewhat higher than we expected.
Point 1: You must check carefully the English language throughout the manuscript.
Response to point 1: The textual parts of the manuscript were reviewed repeatedly, and the English translation was corrected in many places. We tried to use the grammar formulas used in scientific papers.
We would like to especially thank you for the many comments, clarifications and suggestions for corrections related to English grammar and English translation. We believe that with these modifications, the "Englishness" of the manuscript has risen to a higher level.
Thank you very much for the review. We tried to answer all reviewers' questions and correct the manuscript according to the review.

Round 2
Reviewer 1 Report
Comments and Suggestions for Authors
The authors regarded most comments.
Comments:
Line 90-92: The information given is incomplete and does not say which traits aim to improve fertility in heifers and cows.
Please work out this issue that the reader is convinced on the add on value of CI in the total merit index.
The authors should explain the total merit index for Holsteins in Hungary and then explain why it is important to include CI in this total merit index.
EDP: is not an EBV because you do not regress on the index. Why do not you derive the b-values to get an EBV for a simple sire model without further relationships.
Nevertheless, this approach is outdated and should be deleted. The other problem with this model arises from the variance estimates. So, you need to estimate genetic parameters in this simple model for all traits in the total merit index.
Formula (1) why you write y with hat, should be the observed value not an estimate. Sire is random, but symbol equals a fixed effect.
Do not see how do you derive sa from the sire model. What is your expectation for the sire variance (classical model Henderson III). I do not know what SPSS calculates.
Line 162: pe of the cow
Formula 5: please apply the regression on the expected EBV. 2 x EDP overestimates EBV, particularly when numbers of daughters are low.
estimated BV >> EBV, if this is really an EBV and not an average on daughters, corrected for the population mean
Line 234-235: this conclusion is not justified given the data presented.
First, are the data representative for all Hungarian Friesian cows when using data from 6 herds.
You should show model calculations based in selection index theory whether CI improves fertility and total merit index. This means you have to estimate the covariance estimates between CI and all traits in the total merit index. Here you have to use an animal model. Then you can calculate the expected selection response for the single traits and the total merit index.
Author Response
|
The line numbers indicated in the response refer to the corrected version of the manuscript. Comment 1: Line 90-92: The information given is incomplete and does not say which traits aim to improve fertility in heifers and cows. Please work out this issue that the reader is convinced on the add on value of CI in the total merit index. The authors should explain the total merit index for Holsteins in Hungary and then explain why it is important to include CI in this total merit index. Response 1: The aim of this manuscript is not to present, evaluate or expand the Hungarian Holstein-Friesian selection index. The calving interval presented in the manuscript is a trait that is not included in the selection index, so we have less information about it. That is why we chose to evaluate the calving interval as the aim of our study. The calving interval is a very good indicator of the reproductive status of a dairy cattle herd. Therefore, we think that the inclusion of the trait in the selection index can be professionally justified. The transformation and development of the index goes far beyond the scope and goals of this manuscript. Even in the best case, we only formulated one recommendation for the inclusion of the calving interval in the selection index. Comment 2: EDP: is not an EBV because you do not regress on the index. Why do not you derive the b-values to get an EBV for a simple sire model without further relationships. Nevertheless, this approach is outdated and should be deleted. The other problem with this model arises from the variance estimates. So, you need to estimate genetic parameters in this simple model for all traits in the total merit index. Response 2: Neither EDP nor EBV are mentioned in the manuscript. The EPD (Estimated Progeny Difference) is included in the manuscript, which has nothing to do with the selection index or the EBV. The process of EPD calculation is described in detail in the material and method chapter (line 175-180). The results presented in this manuscript have nothing to do with the selection index or the traits included in it. Comment 3: Formula (1) why you write y with hat, should be the observed value not an estimate. Sire is random, but symbol equals a fixed effect. Do not see how do you derive sa from the sire model. What is your expectation for the sire variance (classical model Henderson III). I do not know what SPSS calculates. Response 3: The formula (1) was written according to the editor's suggestion even before the reviews were completed. In this model, the "y" is estimated value. In the explanation below the formula, we indicated that the sire was a random effect and the others were fixed effects. We thought that the method of calculating the variance components is quite well known in the literature, so we did not detail it in the manuscript. The calculation was made based on the data of the ANOVA table: σ2a = 4x [(MSsire - MSE) / k1)] σ2e = MSE σ2p = σ2a + σ2e In SPSS: General Linear Model / Variance Components
Comment 4: Line 162: pe of the cow Response 4: Thank you very much, we have corrected it.
Comment 5: Formula 5: please apply the regression on the expected EBV. 2 x EDP overestimates EBV, particularly when numbers of daughters are low. estimated BV >> EBV, if this is really an EBV and not an average on daughters, corrected for the population mean Response 5: See response 2.
Comment 6: Line 234-235: this conclusion is not justified given the data presented. Response 6: Did you mean line 329-330? The sentence has been modified: „Based on our results, it seems worth considering the possibility of incorporating CI into the selection index.”
Comment 7: First, are the data representative for all Hungarian Friesian cows when using data from 6 herds. Response 7: We did not examine our database in this aspect.
Comment 8: You should show model calculations based in selection index theory whether CI improves fertility and total merit index. This means you have to estimate the covariance estimates between CI and all traits in the total merit index. Here you have to use an animal model. Then you can calculate the expected selection response for the single traits and the total merit index. Response 8: As previously described, index development is not the aim of this manuscript.
Point 1: English language fine. No issues detected. Response to point 1: Thank You very much. |
